# A Novel Mechanical Frequency Tuning Method Based on Mass-Stiffness Decoupling for MEMS Gyroscopes

**DOI:** 10.3390/mi13071052

**Published:** 2022-06-30

**Authors:** Chuanfu Chen, Kai Wu, Kuo Lu, Qingsong Li, Chengxiang Wang, Xuezhong Wu, Beizhen Wang, Dingbang Xiao

**Affiliations:** 1College of Intelligence Science and Technology, National University of Defense Technology, Changsha 410073, China; chenchuanfu@nudt.edu.cn (C.C.); wukai@nudt.edu.cn (K.W.); lukuo13@nudt.edu.cn (K.L.); cx.wangnudt@nudt.edu.cn (C.W.); xzwu@nudt.edu.cn (X.W.); dingbangxiao@nudt.edu.cn (D.X.); 2Hunan MEMS Research Center, Changsha 410073, China; 3Laboratory of Science and Technology on Integrated Logistics Support, National University of Defense Technology, Changsha 410073, China

**Keywords:** ring MEMS gyroscopes, mass-stiffness decoupling, frequency split, femtosecond laser, frequency tuning

## Abstract

MEMS gyroscopes play an important role in inertial navigation measurements, which mainly works in *n* = 2 mode. However, mode matching is the basis for high-precision detection, which can improve the sensitivity, resolution, and signal-to-noise ratio of the gyroscopes. An initial frequency split is inevitably generated during the manufacturing process. There are two methods to eliminate the frequency split and to achieve mode matching for the gyroscopes, which are electrostatic tuning and mechanical trimming, respectively. In this paper, we report a novel ring MEMS resonator and a novel method of mechanical frequency tuning. The most prominent characteristic of the resonator is that 16 raised mass blocks are increased in the circumferential positions of the ring uniformly. This structural design can achieve mass-stiffness decoupling, which means that punching holes on the mass blocks only affects the mass distribution but the stiffness is almost unchanged for the resonator. We verify the mass-stiffness decoupling by way of comparing the simulation with the conventional resonator. In addition, we put up an online tuning platform based on a femtosecond laser and reduce a resonator’s frequency split from 23.3 Hz to 0.4 Hz, which reveals that the frequency split is linearly related to the removed mass. These findings will have a referential significance for other transducers.

## 1. Introduction

MEMS gyroscopes show great superiority over conventional mechanical gyroscopes in size, power consumption, and cost. Ring MEMS gyroscopes have excellent performance advantages from their symmetrical structure and work in the case of mode-matching [1]. Mode-matching is crucial for the ring resonator, which influences the sensitivity, resolution, and signal-to-noise ratio of the gyroscope [2].

There are two methods to realize mode matching: electrostatic tuning and mechanical trimming. In electrostatic tuning, the DC voltage is usually applied to the electrodes, which decreases the frequency of the corresponding mode of the gyroscope due to the effect of electrostatic negative stiffness; thus, the mode matching is realized. However, it does not change the structure of the resonator in essence and is very sensitive to the environmental temperature. By contrast, mechanical trimming can decrease various errors by changing the mass or stiffness distribution of the resonator. Thus it has been widely applied to high-performance gyroscopes such as micro-hemispherical or cylindrical gyroscopes in recent years.

In previous studies, the mechanical trimming mainly had two methods. However, no matter whether mass is added to the structure or reduced, there are many uncertainties. For example, change of the structure will affect the mass and stiffness at the same time, making it difficult to control the frequency split. In addition, the laser ablation cannot guarantee the shape of punching holes [3,4], and it also leads to stress concentration, which results in the attenuation of the quality factor [5]. Moreover, a directional lapping technique can decrease the frequency split well but it is complicated to determine the position of the rigid-axis [6]. Although adding mass has little effect on the quality factor, it requires high-precision equipment and is high-cost, which creates many difficulties for frequency tuning [7,8,9].

To improve the performance of the Disk Resonant Gyro, it is verified that hanging mass blocks on the ring can maintain the stiffness of the resonator [10]. In this paper, we come up with a new ring resonator that can achieve the decoupling between mass and stiffness when changing the mass on the structure. Based on this structure, we put forward a novel method of permanent frequency split reduction by laser ablation. It not only increases the convenience of frequency tuning but also lowers the damage to the structure caused by the laser, which can reduce the decline of the quality factor for ring MEMS gyroscopes.

## 2. Model Design for the New Ring Resonator

### 2.1. Model Design

The traditional ring MEMS resonant structure is composed of a support beam and a resonant ring. Any change of mass at any location on the ring will affect the mass and stiffness distribution of the structure. The overall frame of the new ring MEMS resonant structure is a four-sided anchor (Figure 1a), where along the circular position of the ring 16-raised small mass blocks are mounted evenly. These mass blocks are connected to the resonant ring by a very short thin beam, for which the change of stiffness is negligible when the mass is changed. Therefore, the resonator can realize the effect of mass-stiffness decoupling theoretically. Ideally, the drive mode’s frequency is equal to the sense modes for the ring MEMS resonator, whose modal simulation results are shown in Figure 1b. As can be seen, the two modal rigid axes are at an angle of 45° to each other. Fused silica is used for the resonator to achieve a high-quality factor, and the structure is fabricated by femtosecond laser pulses assisted by with chemical etching.

### 2.2. Theory of Frequency Tuning

To achieve the effect of a ring resonator with errors, suppose that $N_{m}$ mass points (as shown in Figure 2) are added to a perfect resonator whose mass is $M_{0}$. Each mass point is expressed by a mass $m_{i}$ and its corresponding circumferential angle of $\phi_{i}$. It only just considers the uneven distribution of mass here, not the influence of stiffness at all.

According to the above model, we can deduce that the position of the high-frequency rigid axis is satisfied with:(1)$$\tan4\phi =\frac{-\sum_{i=1}^{N_{m}}\left(m_{i}\sin4\varphi_{i}\right)}{-\sum_{i=1}^{N_{m}}\left(m_{i}\cos4\varphi_{i}\right)}$$

The two frequencies of the ring resonator’s operating modes are:(2)$$\Omega_{1}{}^{2}=\Omega_{0}{}^{2}\left\{\frac{(1+\alpha^{2})M_{0}}{(1+\alpha^{2})M-(1-\alpha^{2})\sum_{i=1}^{N_{m}}\left(m_{k}\cos4(\phi_{i}-\varphi )\right)}\right\}$$
(3)$$\Omega_{2}{}^{2}=\Omega_{0}{}^{2}\left\{\frac{(1+\alpha^{2})M_{0}}{(1+\alpha^{2})M+(1-\alpha^{2})\sum_{i=1}^{N_{m}}\left(m_{k}\cos4(\phi_{i}-\varphi )\right)}\right\}$$
where α is the ratio of radial amplitude to tangential amplitude, $\Omega_{0}$ is the resonant frequency of a perfect ring resonator.

Furthermore, the value of frequency split can be deduced as follows:(4)$$\Omega_{1}-\Omega_{2}=-\frac{(1-\alpha^{2})\Omega_{0}}{(1+\alpha^{2})M}\left[\sum_{i=1}^{N_{m}}\cos4\phi_{i}\cos4\varphi +\sum_{i=1}^{N_{m}}\sin4\phi_{i}\sin4\varphi \right]$$

From the mathematical form of $\sin4\phi_{i}$, $\cos4\phi_{i}$, sin4φ, cos4φ, in the Equations from (1) to (4), it can be concluded that if we remove the same mass at an interval of 90°, we can get the equal tuning effect; but if at an interval of 45°, the tuning effect will be canceled out.

## 3. Simulation Analysis

### 3.1. Simulation of Mass-Stiffness Decoupling

Then, we need to compute the resonator’s effective mass and effective stiffness to know whether it has the character of mass-stiffness decoupling. On the one hand, we increase the radius of the mass blocks to observe how the resonator’s effective mass and effective stiffness change (as shown in Figure 3). We can see the effective mass has a large variable range but the effective stiffness is almost unchanged. On the other hand, we punch 16 holes of which the diameter is from 100 μm to 500 μm, by the step of 100 μm symmetrically on the ring and mass blocks, respectively (as shown in Figure 4). We can draw up the curves of effective stiffness, effective mass and histograms of their growth rates as shown in Figure 5. The left shows punching holes on the ring and the right is punching holes on the mass blocks. However, we have to take overall considerations to the shock impact and the modal frequency of mass blocks. So we can choose a better set of parameters of the mass blocks.

If punching holes on the ring, the resonator’s effective mass and effective stiffness will change simultaneously, which means the relationship of them does not decouple. However, if punching holes on the mass blocks, the effective mass has higher growth rates than the effective stiffness. In addition, we can see the effective stiffness is reducing with the punching holes’ depths but the effective mass is nearly unchanged.

### 3.2. Modal Analysis for the New Ring Resonator with Initial Errors

Because of the processing defects, the manufactured resonators will have an initial frequency split. Using the COMSOL simulation software by removing partial mass on some mass blocks will introduce the frequency split. For example, punching a hole (diameter D = 200 μm, depth d = 200 μm) on one mass block will result in uneven mass distribution (as shown in Figure 6a No. 13). The simulation results are shown in Figure 6b. The drive mode’s frequency is 23,159.9 Hz and the sense mode’s is 23,200.3 Hz, so that the initial frequency split is 40.4 Hz.

### 3.3. The Effect of Trimming at 45° and 90° Intervals, Respectively

According to the derivation of the tuning theory above: if removing the same mass at an interval of 90°, the tuning effects are equal to each other; but at an interval of 45°, the tuning effect will be canceled out. We assume that punching holes whose diameters are 100 μm and depths are 200 μm on the mass blocks of the resonator with the errors shown in Figure 5. These holes are situated on the mass blocks numbered 11, 15, 3, and 7, respectively (only punch one hole at a time), and the simulation results are shown in Table 1. During the simulation, punching holes’ positions are all in the center of the mass blocks. It must be pointed out that the removed mass is ideally consistent with that set in the software, which provides significant guidance for the actual trimming.

From Table 1 above, we can see that punching holes at 90° intervals introduces the same frequency split, that is, the positions at 90° intervals on the resonator are equivalent. The two frequencies are also increasing.

Once more, different from the punching set out in Table 1, we assume punching the same holes continuously on the mass blocks for which the order is numbers 11, 15, 3, 7, respectively. We record the frequency split during the process. Simulation results are shown in Table 2.

From Table 2, we can see the frequency split decrease gradually and approach close to 0. On the one hand, each punching of a hole will reduce the frequency split by about 10 Hz. Thus, we punch holes on the right mass blocks which are close to the low frequency axis. On the other hand, if we punch two holes continuously at 90° intervals, the tuning effect is doubled compared to punching one hole.

Afterwards, we may simulate punching holes of diameter D = 150 μm, depth d = 300 μm sequentially on the resonator’s mass blocks with the errors shown in Figure 5 numbered 1–4, and with the number sequence 1, 3, 2, and 4. Similarly, we record the intermediate process in terms of the changes of frequency split. The simulation results are shown in Table 3.

From the above Table 3, we can see the frequency split increases or decreases based on timing. For example, the first punching hole makes the initial frequency split decrease about 33.8 Hz, but after punching the second hole, the frequency split goes back to the origin. Similarly, the effect of punching the third and the fourth hole is the same as before. However, the number 1 mass block is at 45° to the number 3 and the number 2 is at 45° tos number 4. Therefore, we can deduce that punching the same holes on the mass blockswhich are at 45° each other, the frequency split will not change. That is to say, the effect of frequency tuning cancels out mutually.

### 3.4. The Relationship between Hole’s Sizes and Frequency Split

In order to study the relationship between frequency split and the mass removed, now, we simulate punching holes on the ring and mass block with various diameters and depths, and then observe and record the frequency split during the punching. We take the diameters from 50 μm to 120 μm by steps of 10 μm and the depths from 50 μm to 400 μm by steps of 50 μm. We only consider punching holes on the stationary position so that we do not change the position of the rigid axis during the punching. The simulation results of punching holes on the ring and mass blocks are shown in Figure 7.

As shown in Figure 7a, punching holes on the ring shows that the curves are not fixed by a law, since there are some curves going up gradually with the depths or diameters increasing but the others are descending. However, the biggest feature of the graph is that the range of the frequency split’s change is about 15 Hz which means if a resonator’s initial frequency split is over 15 Hz, punching holes on the ring will not be able to tune it. However, punching holes on the mass blocks will have obvious results, as shown in Figure 7b. With the depths and diameters rising, the frequency split grows and the relationship between them is perfect linearity. Moreover, the range of frequency split depends on how much mass to remove. In general, removing one mass block completely can reduce the split by about 160 Hz at most. As a matter of fact, according to the formula of frequency, stiffness and mass, punching holes on the ring will not only affect the structural stiffness but also mass distribution, so the frequency split’s curves have no fixed laws. Thus, punching holes on the mass blocks almost only affects the mass distribution of the resonator rather than stiffness. Hence, as the mass is increasing and stiffness is unchanged, the frequency split will increase gradually.

## 4. Experimental Verification and Analysis of Results

### 4.1. Tuning Platform and Actual Punching Effect

Finally, we build an online tuning platform based on femtosecond laser (as shown in Figure 8). The platform is mainly composed of two parts: one is for tuning of the execution system based on femtosecond laser, which includes laser, light speed conversion system, galvanometer, miniature vacuum chamber and carrier platform; the other is a feedback system of frequency sweep, which includes a working computer (or a laptop), a lock-in amplifier and a DC power supply. The laser repetition rate is 200 kHz, single pulse energy is 20 W, jump speed is 150 mm/s and the processing speed is 120 mm/s.

The whole experimental process is: use lock-in amplifier to sweep the gyroscope’s modal frequencies, according to the size of the frequency split then operate the femtosecond laser to punch holes on the mass blocks. After the end of the punching, sweep the frequency again to observe how the frequency split changed. We determine the next punching position and hole size by frequency split and relative amplitude of two peaks of the curve. Punch holes without a break until the frequency split meets the requirement.

The gyro is separated from the test circuit board to avoid the influence of temperature, which will result in drifting in the angular frequency and lead to noise. As the laser parameters will have an impact on punching holes, such as diameter, depth and so on, we try punching holes on the dilapidated resonator for exploring which parameter is the best before the tuning experiment. The spot diameter of the laser is about 7–10 μm; we punch two holes of diameters 60 μm and 100 μm on the mass blocks, but we can see the hole’s diameters are about 70 μm and 110 μm under the electron microscope as shown in Figure 9a, which is inevitable. Then, the second graph (Figure 9b) below shows the sectional view of the punched holes, and we can see the diameters are gradually decreasing from top to bottom. In addition, the laser punching does not completely match the set value, and the actual punching shape is conical instead of cylindrical.

### 4.2. Tuning Experiment

After all the simulations above, it is so important to verify if these simulation or theories are consistent with the experiment. Firstly, we punch holes of diameter 100 μm and depth 150 μm on the mass blocks of a resonator with an initial frequency split which is 15.7 Hz under vacuum. The punching sequence of the mass blocks are numbers 13, 15, 11 and 9 which are at 90° intervals. However, in the actual punching, the holes’ centers do not completely coincide with the mass blocks. The results are shown in Table 4.

From the results above, we can see the modal frequencies are all rising after punching, which is consistent with the simulation. Likewise, punching positions at 45° intervals cancels out the tuning effect mutually. Then, we continue punching holes on the mass blocks numbered 2, 6, 8 and 12 in order, whose diameters are 60 μm and depths are 150 μm. The results are shown in Table 5.

As we can see in Table 5, the frequency split is decreasing slowly; it reveals that the punching position is close to the low-frequency rigid axis and the removed mass is less, so that the frequency split is almost unchanged. These mass blocks are at 90° intervals, and every punch will result in about 1 Hz. We can deduce that the tuning positions at 90° intervals are equivalent to each other, in other words, the tuning effect will be multifold. These results show that the rigid axes are at 45° intervals for the *n* = 2 mode of ring resonators.

Next, we begin to punch holes on another resonator under vacuum to achieve mode matching, of which the frequency sweep is shown as Figure 10a. Its modal frequencies are 24,597.3 Hz and 24,620.6 Hz, respectively, so the initial frequency split is 23.3 Hz.

As shown in Figure 10a, the frequency sweep curve has two peaks which indicates that the rigid axis position of the resonator is not fully aligned with the driving force direction. We need to punch holes on the resonator for sake of diminishing the frequency split. Now, we number the resonator’s mass blocks in the way shown in Figure 5. The first set of experiments is punching holes on the mass blocks numbered 5, 1, 9 and 13 in sequence, of which the diameters are 60 μm and the depths are 500 μm. We record the frequency split after every time of punching so that we can decide which mass block is close or far to the low-frequency rigid axis. The first set of experiments ends up with the frequency split 31.2 Hz and the resonator’s modal frequencies are 24,609.9 Hz and 24,641.1 Hz, respectively, as shown in Figure 10b. According to the above simulation, the tuning effect is equivalent for the mass blocks which are at 90° intervals. We can see the frequency split increase and the modal frequencies are also increasing gradually; therefore, we have punched the mass blocks far from low-frequency rigid axis.

On the basis of the first set of experiments, the second group of experiments goes on punching holes on the mass blocks numbered 12, 8, 16 and 4 sequentially. These holes’ diameters and depths are 60 μm and 500 μm respectively. As shown in Figure 10c, the resonator’s modal frequencies are 24,614.0 Hz and 24,652.4 Hz, and its frequency split is 38.4 Hz. We can see the frequency split has a further increase, which means the high-frequency rigid axis is between the mass blocks numbered 1 and 16, so we can judge that the low-frequency rigid axis is between number 2 and number 3. Comparing the Figure 10b,c, we can see the relative amplitude represent the defection angle of the rigid axis. The bigger the defection angle is, the smaller the relative amplitude among the frequency sweep’s two peaks.

On the basis of the second set of experiments, the third group of experiments continues to punch holes on the mass blocks numbered 2, 6, 10 and 14 in turn. From Figure 10d, we can see the frequency split decreases. By means of the frequency sweep, we can acquire that the resonator’s modal frequencies are 24,635.8 Hz and 24,662.3 Hz, respectively, and the frequency split is 26.5 Hz.

On the basis of the third set of experiments, finally the fourth group of experiments is designed, in which we punch holes on the mass blocks numbered 15, 11 and 7 in order, of which the diameters are 100 μm and the depths are from 200 μm to 400 μm by the step of 100 μm. At the end of the final punching, the resonator’s modal frequencies are 24,662.3 Hz and 24,662.7 Hz. It is worth mentioning that the final frequency split is 0.4 Hz, as shown in Figure 10e. We can see that the frequency sweep only has one peak which means the rigid axis is aligned with the driving force direction. Thus, if we want to find out what the frequency split is, we must change the electrode for frequency sweep.

The tuning parameters of the four sets of experiments are listed in Table 6. In Figure 11, we draw the curve of punching time and frequency split from the first set of experiments to the fourth. We also give the schematic diagram of the punching positions, the holes’ diameters and depths on the top of Figure 11. It can be obtained that the removed mass is linear to frequency split, approximately, which is because during the punching of the holes, the rigid axis will deflect. Punching holes on the different mass blocks causes the frequency split to increase or decrease, depending on whether the mass block is close to the low-frequency rigid axis.

## 5. Conclusions

We have come up with a new ring resonator which was different from the conventional resonator and could achieve the decoupling between mass and stiffness by punching holes on the mass blocks. Based on this structure, we put forward a novel method of permanent frequency split reduction by laser ablation. It not only increases the convenience of the frequency tuning but also lowers the damage to the structure with the laser ablation, which can reduce the quality factor of the resonator. Moreover, our research will have a referential significance to other transducers.

## Figures and Tables

**Figure 1 micromachines-13-01052-f001:**
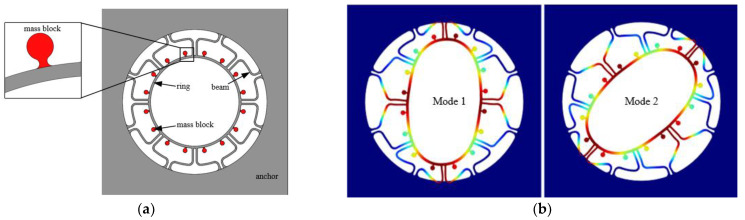
(**a**) New ring resonant structure; (**b**) Simulation of ideal ring resonator.

**Figure 2 micromachines-13-01052-f002:**
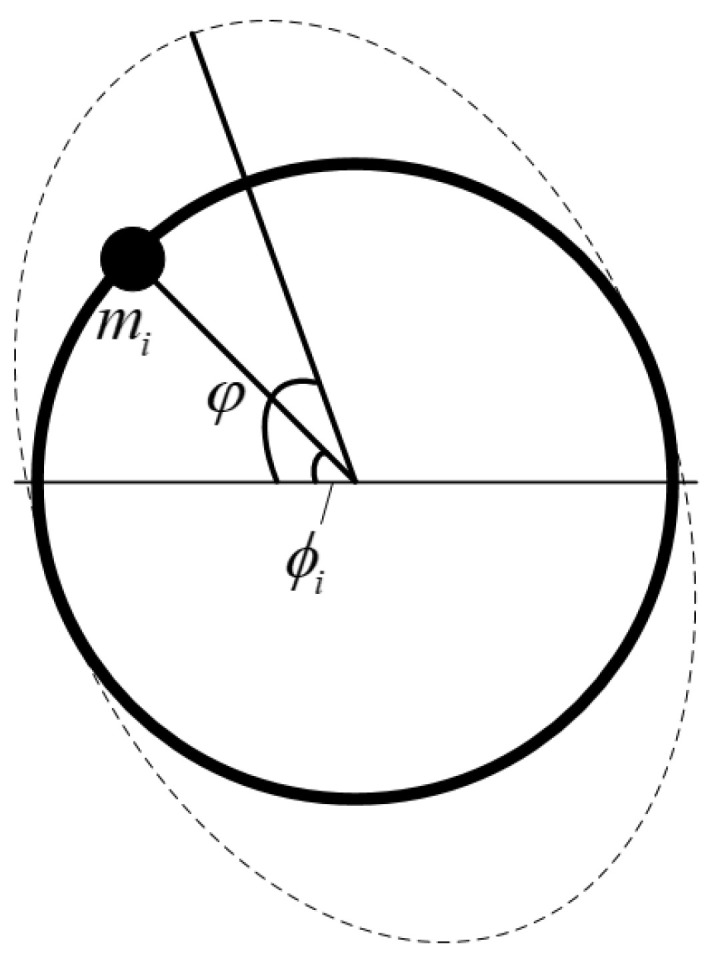
Perfect ring resonator adding mass points.

**Figure 3 micromachines-13-01052-f003:**
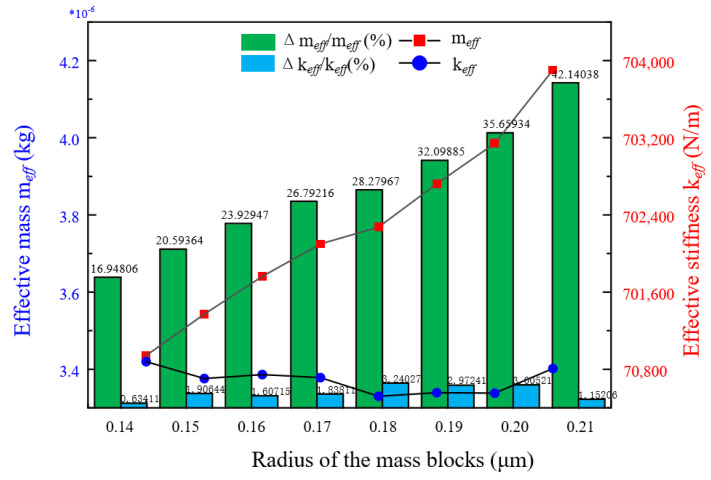
The decoupling effect varies with the radius of mass blocks.

**Figure 4 micromachines-13-01052-f004:**
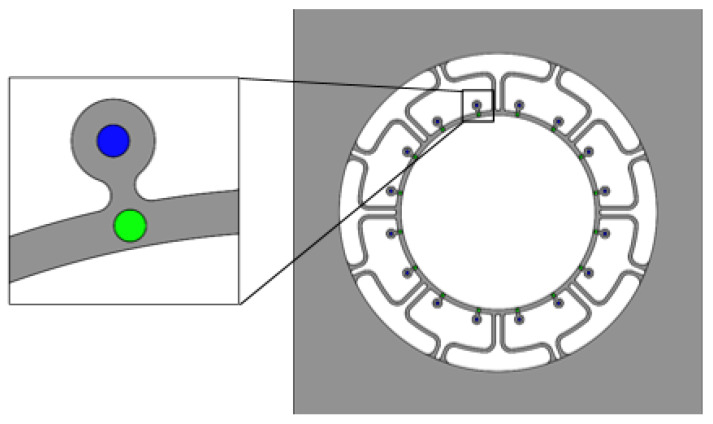
Punch 16 holes evenly on the mass blocks and the ring, respectively.

**Figure 5 micromachines-13-01052-f005:**
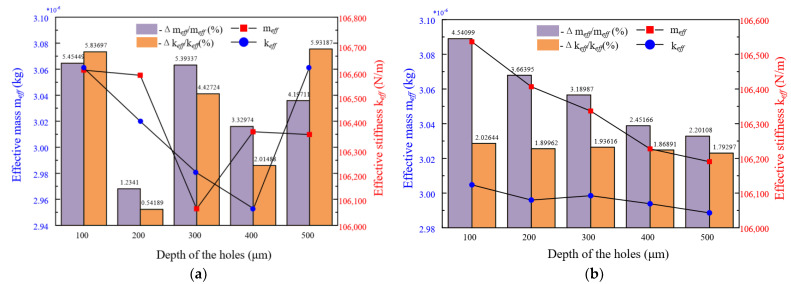
Compare the influence of punching holes at different places for the effective mass and effective stiffness. (**a**) Punch holes on the ring; (**b**) Punch holes on the mass blocks.

**Figure 6 micromachines-13-01052-f006:**
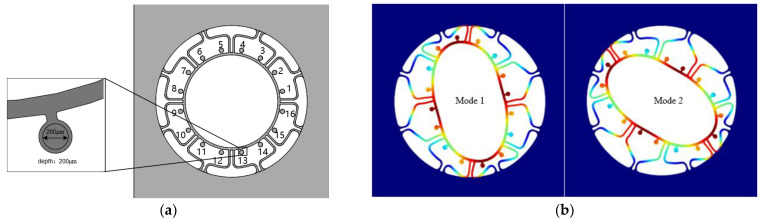
Resonator with processing errors and its modal simulation. (**a**) Resonator with processing errors; (**b**) Modal simulation.

**Figure 7 micromachines-13-01052-f007:**
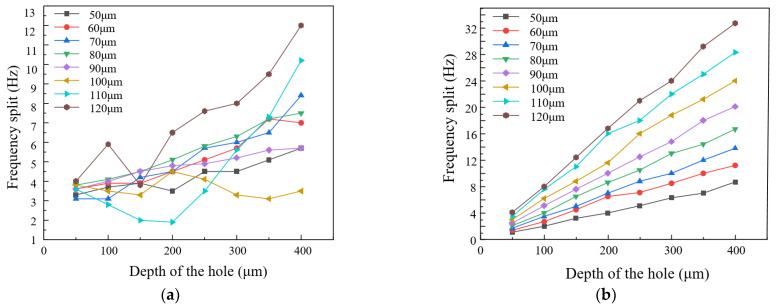
The curves of the relationship between punching sizes and frequency split. (**a**) Punch holes on the ring; (**b**) Punch holes on mass blocks.

**Figure 8 micromachines-13-01052-f008:**
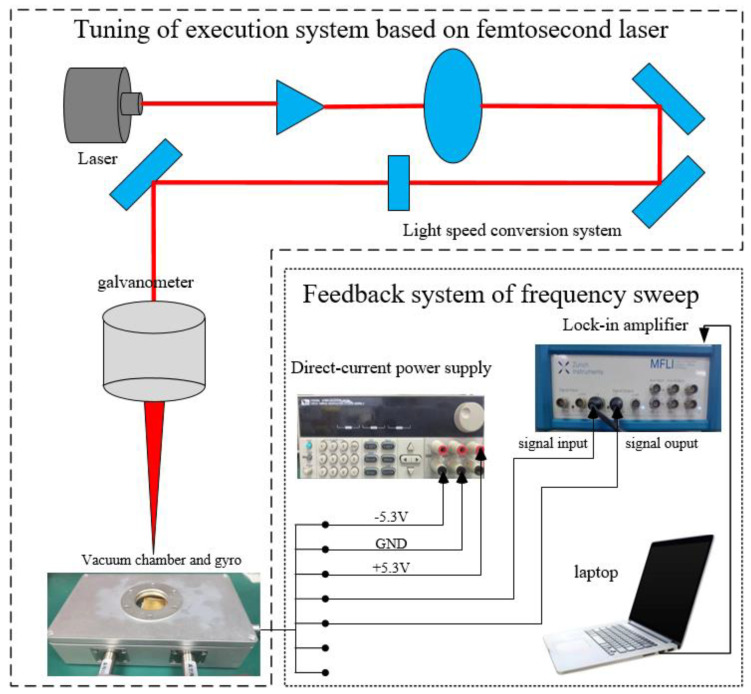
Online tuning platform based on femtosecond laser.

**Figure 9 micromachines-13-01052-f009:**
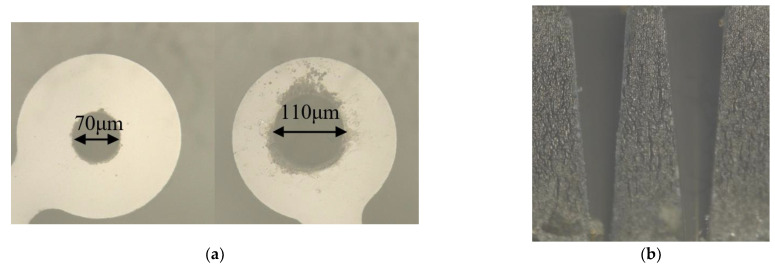
Sketch of the actual punching: (**a**) The diameters are 60 μm and 100 μm; (**b**) Sectional view of holes.

**Figure 10 micromachines-13-01052-f010:**
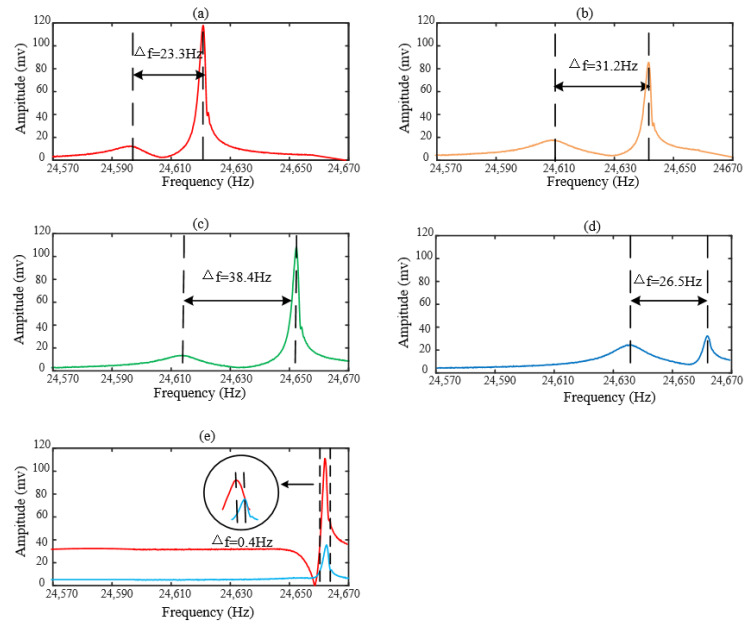
Frequency sweep of tuning process. (**a**) The initial frequency sweep; (**b**) Frequency sweep after the first time of tuning; (**c**) Frequency sweep after the second time of tuning; (**d**) Frequency sweep after the third time of tuning; (**e**) Frequency sweep after the fourth time of tuning.

**Figure 11 micromachines-13-01052-f011:**
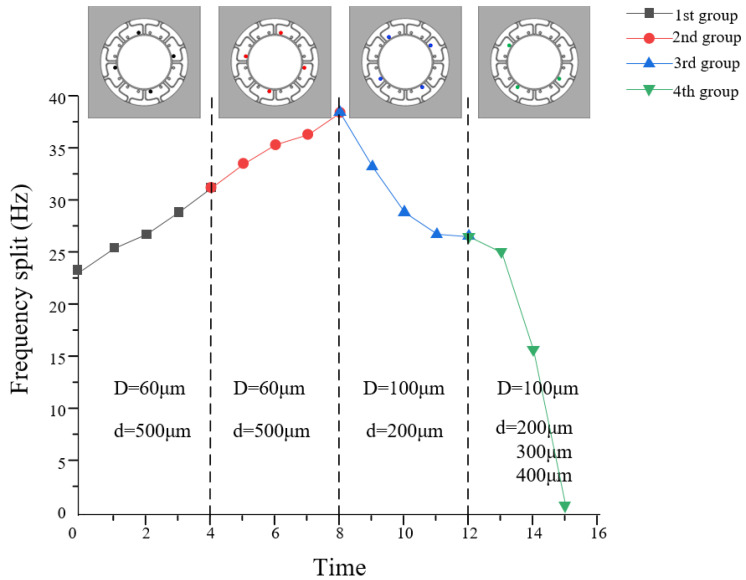
The correlation curves between frequency split and removed mass in four sets of experiments.

**Table 1 micromachines-13-01052-t001:** Record of the frequency split for different tuning positions at 90° intervals.

Number	Frequency 1 (Hz)	Frequency 2 (Hz)	Split (Hz)
Initial	23,159.9	23,200.3	40.4
11	23,170.6	23,201.2	30.6
15	23,170.8	23,201.9	31.1
3	23,170.7	23,201.9	31.2
7	23,170.8	23,201.8	31.0

**Table 2 micromachines-13-01052-t002:** Record of the frequency split for continuous punching holes at 90° intervals.

Order	Frequency 1 (Hz)	Frequency 2 (Hz)	Split (Hz)
11	23,170.6	23,201.2	30.6
15	23,185.6	23,205.7	20.2
3	23,215.7	23,226.1	10.4
7	23,255.2	23,255.3	0.1

**Table 3 micromachines-13-01052-t003:** Record of the frequency split for continuous punching holes at 45° intervals.

Order	Frequency 1 (Hz)	Frequency 2 (Hz)	Split (Hz)
1	23,165.1	23,239.4	74.3
3	23,203.4	23,243.9	40.5
2	23,217.7	23,273.2	54.5
4	23,246.8	23,287.2	40.1

**Table 4 micromachines-13-01052-t004:** Modal frequencies and frequency splits recorded during the tuning process.

Order	Frequency 1 (Hz)	Frequency 2 (Hz)	Split (Hz)
Initial	24,572.5	24,588.2	15.7
13	24,585.6	24,605.3	19.7
15	24,594.2	24,610.1	15.9
11	24,601.0	24,612.4	11.4
9	24,606.5	24,622.5	16.0

**Table 5 micromachines-13-01052-t005:** Modal frequencies and frequency split recorded during the tuning process.

Order	Frequency 1 (Hz)	Frequency 2 (Hz)	Split (Hz)
Initial	24,606.5	24,622.5	16.0
2	24,609.1	24,624.3	15.2
6	24,612.5	24,626.5	14.0
8	24,616.0	24,628.9	12.9
12	24,620.8	24,634.4	11.6

**Table 6 micromachines-13-01052-t006:** Tuning parameters in the experiments.

Experiment	Mass Block	Diameter (μm)	Depth (μm)
1st	5	60	500
	1		
	9		
	13		
2nd	12	60	500
	8		
	16		
	14		
3rd	2	100	200
	6		
	10		
	14		
4th	15	100	200
	11		300
	7		400

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
