# Peer review of "A Novel Mechanical Frequency Tuning Method Based on Mass-Stiffness Decoupling for MEMS Gyroscopes"

_micromachines, 2022, doi:10.3390/mi13071052_

Round 1
Reviewer 1 Report
Dear Authors,
I like the general approach with numerical and experimental analysis However, there are some concerns that need to be addressed. They are listed as follows:
o The introduction of the paper is too general to give a clear explanation of why the research of the thesis is carried out, and its inherent logic and innovation are not outstanding.
o How did you correct any drifting in the angular frequency? This in fact leads to noise.
o It looks that the mass-stiffness decoupling process is an optimization problem. Am I correct?
o It is realistic and possible to fabricate mass blocks. We know that there are several uncertainties during fabrication. This will impact the Gyroscope sensitivity by default. How do you plan to overcome this problem?
o There is no big difference in modes 1 and 2. A split frequency of 31 Hz is not enough to decouple the two modes from a structural dynamic point of view.
o Line 87 is not clear, where are the mathematical form of sin 4 \phi and the rest?
o Please use the MDPI template. The current format has several issues. For example, the text style and line spacing are not consistent. Compare lines 111-119 to that from 121-127.
o Which fabrication process is used? Add details.
o What is the tuning parameter in the experimental test?
o Based on the reported results, which configuration has high sensitivity and why?
o Please rewrite the conclusion and focus on the findings of this work.
o Several typos need to be fixed.
Reviewer 2 Report
The manuscript proposed a method to tune mechanical frequency for MEMS gyroscopes based on mass-stiffness decoupling. There are some comments and suggestions:
1. What is your consideration of the number of small mass blocks in your design are 16? How about less or more than 16 mass blocks?
2. I suggest making a table to describe your four sets of experiments and their different conditions to make the reader easy to understand.
3. Based on the 1st and 2nd experiments, why do the different punching sequences of the mass blocks give different results?
Round 2
Reviewer 1 Report
Dear Authors,
Thanks for the report. I have revised the updated version and it looks great. However, I still believe a further modification is needed. The introduction is still not suitable and does not cover the related works perfectly. It should state the main contribution and the novelty compared to the previous works. The English language needs a major revision. Mathematical modeling requires a further explanation.